# Molecular Cytogenetic Identification of Wheat-*Aegilops Biuncialis* 5M^b^ Disomic Addition Line with Tenacious and Black Glumes

**DOI:** 10.3390/ijms21114053

**Published:** 2020-06-05

**Authors:** Liqiang Song, Hui Zhao, Zhi Zhang, Shuai Zhang, Jiajia Liu, Wei Zhang, Na Zhang, Jun Ji, Lihui Li, Junming Li

**Affiliations:** 1Center for Agricultural Resources Research, Institute of Genetics and Developmental Biology, The Innovative Academy of Seed Design, Chinese Academy of Sciences, Shijiazhuang 050022, China; lqsong@sjziam.ac.cn (L.S.); w_as_d@163.com (S.Z.); ljjzl2014@163.com (J.L.); edithor@126.com (W.Z.); zhangna2nina@163.com (N.Z.); jijun@siziam.ac.cn (J.J.); 2State Key Laboratory of Plant Cell and Chromosomal Engineering, Chinese Academy of Sciences, Beijing 100101, China; 3College of Bioscience and Bioengineering, Hebei University of Science and Technology, Shijiazhuang 050018, China; applecong@126.com; 4Institute of Crop Science, Chinese Academy of Agricultural Sciences, Beijing 100081, China; zhangzhihkd@126.com (Z.Z.); lilihui@caas.cn (L.L.)

**Keywords:** common wheat, *Aegilops biuncialis*, 5M^b^, SLAF markers, glume trait

## Abstract

Production of wheat-alien disomic addition lines is of great value to the exploitation and utilization of elite genes originated from related species to wheat. In this study, a novel wheat-*Aegilops biuncialis* 5M^b^ disomic addition line WA317 was characterized by in situ hybridization (ISH) and specific-locus amplified fragment sequencing (SLAF-seq) markers. Compared to its parent Chinese Spring (CS), the glumes of WA317 had black color and were difficult to remove after harvesting, suggesting chromosome 5M^b^ carried gene(s) related to glume development and *Triticeae* domestication process. A total of 242 *Ae. biuncialis* SLAF-based markers (298 amplified patterns) were developed and further divided into four categories by *Ae. biuncialis* Y17, *Ae. umbellulata* Y139 and *Ae. comosa* Y258, including 172 markers amplifying the same bands of U and M genome, six and 102 markers amplifying U-specific and M-specific bands, respectively and eighteen markers amplifying specific bands in Y17. Among them, 45 markers had the specific amplifications in WA317 and were 5M^b^ specific markers. Taken together, line WA317 with tenacious and black glumes should serve as the foundation for understanding of the *Triticeae* domestication process and further exploitation of primitive alleles for wheat improvement. *Ae. biuncialis* SLAF-based markers can be used for studying syntenic relationships between U and M genomes as well as rapid tracking of U and M chromosomal segments in wheat background.

## 1. Introduction

Transferring desirable genes from wild relatives into common wheat is considered an efficient strategy for wheat genetic improvement. Wheat alien disomic addition lines are usually used as intermediate materials to further produce alien translocation and introgression lines for wheat breeding. On the other hand, alien disomic addition lines are commonly used for sorting alien chromosomes [1], homology comparison between wheat and alien relatives [2], chromosomal localization of alien genes [3] and development of molecular markers [4]. To date, a large number of wheat–alien chromosome addition lines and amphiploids/introgression lines have been obtained from crossing with wild species, including *Secale cereale* [5], *Haynaldia villosa* [6], *Thinopyrum ponticum* [7], *Aegilops speltoides* [8] and *Agropyron cristatum* [9].

Cytogenetic techniques have always been an important methodology of detecting wild relative introgressions [10,11]. New experimental protocols [12], oligonucleotide probes [13] and other innovations improved the procedures of the preparation of root tip cells and detecting probes, as well as the process of in situ hybridization. Despite this, molecular markers are considerably more popular in identifying alien chromosomes because of their high efficiency and throughput. Simple sequence repeats (SSR) [14,15], expressed sequence tag (EST) [16], sequence-tagged sites (STS) [17], cleaved amplification polymorphism sequence (CAPS) [18] and competitive allele-specific PCR (KASP) [19] markers have been successfully used to identify alien derivative lines in large numbers. Specific-locus amplified fragment sequencing (SLAF)-seq technology has several obvious advantages, including high throughput, high accuracy and low cost and provides an important tool for developing specific markers of wheat wild relatives, such as *Thinopyrum elongatum* 7E [20], *Agropyron cristatum* [21], *Thinopyrum ponticum* 4Ag [22], rye 4R ^Ku^ [23] and 6R ^Ku^ [24] chromosomes. These SLAF-based markers could be used for tracking specific alien chromosomal segments and alien genes in wheat background. To date, no *Ae. biuncialis* specific SLAF-based markers have been reported.

Modern common wheat varieties differ from their wild ancestors due to the acquisition of domestication traits. Discovering genes governing these traits will provide more insight into the domestication process and lead to further exploitation of primitive alleles for wheat improvement [25,26]. Threshability is mainly controlled by two genes—the major domestication gene *Q* and the homoeologous tenacious glume (*Tg*) genes. The *Q* gene on chromosome arm 5AL, a member of the APETALA2 (*AP2*) family of transcription factors, affects multiple agronomic traits, such as threshability, rachis fragility, plant height, spike length and glume morphology. Sharma et al. [27] mapped these major quantitative trait loci (QTL) loci through a population of recombinant inbred lines derived from a cross between durum and cultivated emmer. *Tg* gene was located on the homoeologous group 2, such as 2A in spelt [28], 2B in wild emmer [29], 2D in *Ae. tauschii* [30,31,32] and 2E in *Lophopyrum elongatum* [33]. Zhang et al. [34] demonstrated that *Q* governs threshability through extensive modification of wheat glumes including their structure, cell wall thickness and chemical composition. The glume color is an important taxonomic discriminator in wheat [35,36] and is associated with the crop’s adaptability [37,38]. The genes controlling the glume color are mainly located on the homoeologous group 1, including *Rg-A1*, *Rg-B1* and *Rg-D1* in common wheat and its related species [39]. Four alleles of *Rg-A1* determine the red color (*Rg-A1b*), the black color (*Rg-A1c* and *Rg-A1d*) and the absence of color (*Rg-A1a*). *Rg-B1b* and *Rg-D1b* control the red color of glumes, while *Rg-B1a* and *Rg-D1a* do not impart color. The *Rg* gene expression in glumes is involved in the biosynthesis of flavonoid pigments phlobaphenes and/or 3-desoxyanthocyanidines [40]. In addition to the red and black glumes, smoky-grey glumes [41] and purple glumes [42] have been also mapped on chromosome 1D of *Triticum aestivum* and chromosome 2A of *Triticum durum*, respectively. A deeper comprehension of the genetic basis of glume traits (threshability and color) will be beneficial to understanding spike developmental mechanism in *Triticeae* species.

*Aegilops* genus comprises 11 diploid, 10 tetraploid and two hexaploid species [43] with various genomic compositions of the C, D, N, M, S and U genomes [44]. Most *Aegilops* species have been successfully introgressed into common wheat. *Ae. biuncialis* (2n = 4x = 28, UUMM) displays a series of agronomically useful traits including special high molecular weight glutenin subunits, disease resistance and drought and salt tolerance [45,46,47]. Molnár-Láng team reported seven wheat–*Ae. biuncialis* disomic addition lines (2U^b^, 2M^b^, 3U^b^, 3M^b^, 5U^b^, 6M^b^ and 7U^b^) in the background of winter wheat line Mv9kr1 [48,49]. During their research, Schneider et al. [50] screened two markers (GWM44 and GDM61) from 108 wheat SSR markers, which amplified specific PCR products from wheat-*Ae. biuncialis* 2M^b^ and 3M^b^ addition lines and Molnár et al. [51] mapped a series of gene-based conserved orthologous set (COS) markers to U and M genomes by wheat-*Aegilops* chromosome introgression lines. Zhou et al. [47] obtained five *Ae. biuncialis* 1U^b^ specific markers from 48 polymerase chain reaction (PCR)-based landmark unique gene (PLUG) markers and created a wheat–*Ae.biuncialis* 1U^b^ disomic addition line using *T. aestivum* cv. Chuannong 19 as receptor parent. To date, most *Ae. biuncialis* specific markers have originated from wheat markers and acquired low efficiency. New *Ae. biuncialis* markers need to be urgently developed to accelerate the process of theoretical research and elite gene utilization in wheat breeding. In this study, a new 5M^b^ addition line derived from the cross between *T. aestivum* cv. Chinese Spring and *Ae. biuncialis* was identified by in situ hybridization (ISH) and SLAF-based markers. Its agronomic performance was also evaluated.

## 2. Results

### 2.1. Genomic In Situ Hybridization (GISH) Analysis

Using *Ae. biuncialis* genome DNA as a probe, GISH detection was performed to verify the presence of *Ae. biuncialis* chromatin in WA317 (Figure 1). The result showed that WA317 had 44 chromosomes, including 42 wheat chromosomes stained blue by DAPI and a pair of intact chromosomes with red hybridization signal, which suggested that the alien chromosomes had successfully been introgressed into Chinese Spring (CS) background (Figure 1a,c). GISH analysis of the offspring plants from five consecutive generations of selfing WA317 further confirmed that it had high cytological stability. Therefore, WA317 was a genetically stable disomic addition line.

### 2.2. Non-Denaturing Fluorescence In Situ Hybridization (ND-FISH) Analysis

Following GISH analysis, ND-FISH analysis with three probes, Oligo-pSc119.2 and Oligo-pAs1.0 (or Oligo-pTa535), was used to characterize the *Ae. biuncialis* and wheat chromosomes (Figure 1). With Oligo-pAs1.0 and Oligo-pSc119.2 as probes, the additional chromosomes had both signals in the terminal end of the short arm (Figure 1b). The same hybridization pattern of Oligo-pSc119.2 was also seen on additional *Ae. biuncialis* chromosomes when labeled with Oligo-pSc119.2 and Oligo-pTa535 (Figure 1d). Meanwhile, the two chromosomes did not bear any Oligo-pTa535 hybridization signals. In addition, the arm ratio (L/S) was also a clue to identify the alien chromosome. In WA317, the arm ratio of *Ae. biuncialis* chromosomes was calculated to be 2.26.

### 2.3. Multicolor GISH (mcGISH) Analysis

To confirm the genomic origin of the alien chromosomes of WA317, mcGISH was performed using *Ae. umbellulata* and *Ae. comosa* genomic DNA as probes (Figure 2). In Figure 2a, mcGISH could clearly distinguish the U^b^ (green) and M^b^ (red) genomes of *Ae. biuncialis*. In line WA317 (Figure 2b), the hybridization signal of alien chromosomes was consistent with that of M^b^ genome, indicating that a pair of M^b^ chromosomes were transferred into the wheat background. Among seven M^b^ chromosomes, the FISH pattern and arm ratio were closer to the characters of *Ae. biuncialis* 5M^b^ chromosome described by Wang et al. [52]. Therefore, WA317 was a wheat-*Ae. biuncialis* 5M^b^ addition line.

### 2.4. Development of Ae. Biuncialis Specific Markers and Molecular Detection of Line WA317

High-throughput sequencing identified a total of 643,071 effective SLAFs for *Ae. biuncialis* Y17. Sequence comparison showed that 9342 sequences had less than 50% homology with CS, which were designed to develop *Ae. biuncialis* specific markers. To date, a total of 600 primer pairs have been designed and amplified from *Ae. biuncialis* and CS. Among them, 242 markers showed *Ae. biuncialis* specific bands with a success rate up to 40.33% (Appendix A).

In order to further map these specific markers into U and/or M genomes, 242 markers were used to amplify DNAs of *Ae. biuncialis* Y17 (UUMM genome), *Ae. umbellulata* Y139 (UU genome), *Ae. comosa* Y258 (MM genome) and the negative control CS. The results showed that the 242 markers could amplify 298 specific bands and be divided into four categories (Table 1, Appendix A, Figure 3). A total of 172 markers had the same amplified bands in Y17, Y139 and Y258, suggesting U and M genomes shared the specific bands (Type 1); six markers had the U-specific amplified bands in Y17 and Y139 but had no corresponding fragments in Y258 (Type 2); 102 markers amplified M-specific diagnostic bands in Y17 and Y258 but did not amplify the corresponding fragments in Y139 (Type 3); eighteen markers amplified the specific bands in Y17 (Type 4).

Using the 242 *Ae. biuncialis* specific markers to identify WA317, it was found that 45 markers showed specific bands in WA317 (Table 1, Appendix A), of which, 37 markers amplified the same patterns in Y17, Y139, Y258 and WA317 (Type 1); two markers (*M6201* and *M90496*) had the same amplified bands in Y17 and WA317, without the corresponding bands in Y139 and Y258 (Type 4). Six markers (*M11259*, *M16669*, *M25302*, *M26841*, *M8262* and *M9106*) amplified the same bands in Y17, Y258 and WA317, excluding Y139 (Type 3) (Figure 4). These results further confirmed that a pair of M^b^ genome chromosomes were added to the CS background. These 45 markers could be used to identify the additional chromosome in WA317.

### 2.5. The Agronomic Traits of Line WA317

The agronomic traits of WA317 were investigated in both 2017–2018 and 2018–2019 seasons. WA317 glumes grew gradually black and became tenacious during the grain-filling stage, compared to CS (Figure 5). As a result, the glumes of WA317 were difficult to remove after harvesting. In its parents, the glumes of *Ae. biuncialis* were tenacious and difficult to remove (Appendix A), while it was much easier to remove the glumes of CS. These results suggested that the 5M^b^ chromosome carried gene(s) related to glume development.

The agronomic traits, including plant height, grain number per spike and thousand grain weight of WA317, were significantly different from its receptor parent CS, due to the alien addition of a pair of intact chromosomes (Table 2).

## 3. Discussion

The production of wheat-*Ae. biuncialis* disomic addition lines enables the study of the genetic effects of individual alien chromosomes in common wheat. In previous studies, eight wheat-*Ae. biuncialis* disomic addition lines (1U^b^, 2U^b^, 2M^b^, 3U^b^, 3M^b^, 5U^b^, 6M^b^, 6U^b^ and 7U^b^) were created in different wheat backgrounds [47,48,49,53]. These addition lines showed distinct agronomic traits, such as seed storage protein profile [47], disease resistance [54] and drought tolerance [53], which were of great importance to discovering genes carried by each *Ae. biuncialis* chromosome.

In this study, a new wheat-*Ae. biuncialis* 5M^b^ addition line was identified in the background of Chinese Spring based on in situ hybridization and SLAF markers. mcGISH enables the parental genomes to be discriminated in allopolyploid plants using different total genomic DNA as probes [55]. Molnár and Molnár-láng [56] clearly discriminated the U^b^ and M^b^ chromosomes in *Ae. biuncialis* and wheat-*Ae. biuncialis* amphiploids using the total genomic DNA of *Ae. umbellulata* and *Ae. comosa* as U and M genomic probes. In this paper, mcGISH showed that a pair of M^b^ chromosomes were introgressed into the CS background. This result was also confirmed by SLAF markers. Six markers (*M11259*, *M16669*, *M25302*, *M26841*, *M8262* and *M9106*) could amplify M specific bands in Y17, Y258 and WA317 (Figure 4). The karyotypes of U and M genomes from *Ae. biuncialis* accessions could discriminate different chromosomes based on FISH probes [52]. The arm ratio was also a clue for identifying alien additional chromosomes. The additional chromosome in WA317 had both signals of pAs1.0 and pSc119.2 in the short arm. From 1M^b^ to 7M^b^ chromosomes, only the 5M^b^ short arm had both signals of pAs1.0 and pSc119.2 [52]. In addition, the L/S arm ratio of the alien chromosome was 2.26, which was near the result of 5M^b^ ratio described by Wang et al. [52]. Based on the above results, we concluded that the additional chromosome was 5M^b^. The alien chromosome in WA317 had no obvious signal on the long arm, while the 5M^b^ long arm had the pAs1.0 signal in Wang et al. [52]. This difference might be the result of the genetic diversity of *Ae. biuncialis* accessions or other unknown events.

This addition line WA317 had obvious tenacious glumes, suggesting that *Ae. biuncialis* 5M^b^ chromosome carries gene(s) functioning like gene *Q* which is located on wheat chromosome 5A. Interestingly, the glume of wheat-*Ae. biuncialis* 5M^b^ addition line was significantly black compared to the receptor parent CS, implying that chromosome 5M^b^ would carry glume color related gene(s). In *Triticeae* species, most glume color genes were reported to locate on homoeologous group 1 [38]. Chromosome 1M^g^ of *Ae. geniculata* had a black glume color gene, wheat-*Ae. geniculata* 1M^g^ (1B) substitution line exhibited a darker color than 1M^g^ (1A) and 1M^g^ (1D) substitution line due to conditional epistasis [57]. We do not know whether the glume color change accompanied with the 5M^b^ addition line was related to the structural rearrangement of M genome during allotetraploid formation; the *Ae. biuncialis* chromosome 5M^b^ addition line could be useful for better understanding of the domestication process of *Triticeae* species.

*Aegilops* contribute to the evolution of cultivated wheat and are important sources of genes for wheat improvement. There exists wide genetic variation among *Aegilops* accessions. DNA clones (pSc119.2 and pAs1) display different hybridization signals in four *Ae. biuncialis* accessions and its two diploid progenitor species [48]. Molnár et al. [51,58] compared the syntenic relationships between U and M genomes among diploid (*Ae. umbellulata* and *Ae. comosa*) and allotetraploid (*Ae. biuncialis* and *Ae. geniculata*) *Aegilops* by COS markers and revealed more significant chromosome rearrangements in U genome than M genome, suggesting that *Aegilops* 4, 6 and 7 chromosomes have undergone clear structural rearrangements relative to wheat. Therefore, development of specific markers helps in investigation of the genetic constitutions of alien chromosomes, homoeologous relationships with wheat and allopolyploidization-related molecular processes, such as the P genome of *Ae. cristatum* chromosome [59] and E genome of *Thinopyrum elongatum* [60].

In this study, 242 *Ae. biuncialis* SLAF-based markers (298 amplified patterns) were developed and could be potentially used for tracking the U and M chromosomal segments in the wheat background. Among them, 172 markers (71.07%) amplified the same bands in Y17, Y139 and Y258, suggesting a high syntenic relationship between U and M genomes. Eighteen markers amplified the specific bands in *Ae. biuncialis* Y17 compared to *Ae. umbellulata* Y139 and *Ae. comosa* Y258, implying that the chromosome region in which the 18 corresponding markers were located might undergo chromosome rearrangement events during allopolyploidization. Few U and M specific markers were reported before [47,50,51], so these markers developed in this study can enrich U and M genome marker availability, facilitate the detection of U and M genomes in the wheat background and accelerate elite gene utilization in wheat breeding. It should be noted that there was only an *Ae. biuncialis* (Y17), an *Ae. biuncialis* (Y139) and an *Ae. comosa* accession (Y258) involved in this study. If more *Aegilops* accessions containing the U and/or M genome were used to identify by these markers, the results might be different to some extent, due to the genetic diversity and evolutionary chromosome rearrangements. This means that these *Ae. biuncialis* specific markers can serve as molecular tools to analyze the syntenic relationships of *Aegilops* species, which would eventually help deep understanding of U and M genomes. Moreover, a total of 45 markers has the specific amplifications in wheat-*Ae. biuncialis* 5M^b^ addition line, which could be used for tracking 5M^b^ chromosomal segments in the wheat background.

## 4. Materials and Methods

### 4.1. Plant Materials

The plant materials in this study included *T. aestivum* cv. ‘Chinese Spring’ (CS) (AABBDD), *Ae. biuncialis* accession Y17 (2*n* = 4*X* = 28, U^b^U^b^M^b^M^b^), *Ae. umbellulata* accession Y139 (2*n* = 18, UU), *Ae. comosa* accession Y258 (2*n* = 14, MM) and wheat–*Ae. biuncialis* 5M^b^ disomic addition line WA317 (2*n* = 44). The 5M^b^ addition line WA317 was a BC_2_F_8_ line derived from the cross between CS and *Ae. biuncialis* with the former as the recurrent parent. The creation procedure was as follows—Three F_0_ hybrid seeds were produced by distant hybridization of common wheat CS × *Ae. biuncialis*, two F_1_ hybrid plants survived and one F_1_ plant was successively backcrossed twice with CS as the male parent and then selfed. During this process, the plants were bagged to prevent any cross pollination. *Aegilops* accessions Y17, Y139, Y258 were kept at the Institute of Crop Sciences, Chinese Academy of Agricultural Sciences, China.

### 4.2. GISH Analysis

The mitotic chromosome spreads from root tip cells were prepared and observed as described by Liu et al. [61]. GISH was performed for detection of *Ae. biuncialis* chromatin in line WA317 as described by Liu et al. [61]. The *Ae. biuncialis* Y17 and CS genomic DNAs were utilized as probe and block, at a 1:250 ratio, respectively. The Digoxigenin-Nick Translation Mix, anti-digoxigenin-rhodamine (red) were purchased from Roche, Mannheim, Germany. All the images were observed under a Nikon Eclipse E600 (Japan) fluorescence microscope and captured with a CCD camera (Diagnostic Instruments, Inc., Sterling Heights, MI, USA).

### 4.3. Non-Denaturing FISH Analysis

The synthetic oligonucleotides Oligo-pAs1.0 (or Oligo-pTa535) and Oligo-pSc119.2 can be used to distinguish wheat chromosomes [13]. Wang et al. [52] identified an *Ae. biuncialis* accession karyotype formula and FISH pattern using pAs1.0 and pSc119.2 as probe. The arm ratios (L/S) of 5M^b^ chromosome was near 2.23 and the short arm had the hybridization signals of pAs1.0 and pSc119.2. Therefore, these oligonucleotide probes could be used to characterize wheat chromosomes and *Ae. biuncialis* 5M^b^ chromosome. The oligonucleotides were synthesized by Shanghai Invitrogen Biotechnology Co. Ltd. (Shanghai, China), as described by Tang et al. [13] and ND-FISH analysis was described by Fu et al. [62].

In order to estimate the arm ratio (L/S), we compared the relative length of the long arm and short arm of the additional chromosomes. We photographed 5–10 root-cells at mitotic metaphase from WA317 and measured the length of both arms using the software Image J. According to the ND-FISH signals and arm ratio, we determined alien chromosome constitution as described in a previous report [52].

### 4.4. Multicolor GISH (mcGISH) Analysis

The total genomic DNA of *Ae. umbellulata* Y139 (UU) was labeled with fluorescein-12-dUTP and total genomic DNA of *Ae. comosa* Y258 (MM) was labeled with Texas-red-5-dUTP, while total genomic DNA of durum wheat (*Triticum turgidum*, 2*n* = 4*X* = 28, AABB) was used for blocking with a ratio of 1:30 [56]. The hybridization mixture (8 μL per slide) included 50 ng of each U and M genome probe and 1.5 μg competitor DNA in 2× SSC and 1× TE buffer. The denaturation and hybridization conditions were previously described by Han et al. [63].

### 4.5. Development of Ae. Biuncialis Specific Markers and Identification of The Addition Line WA317

SLAF-seq of *Ae. biuncialis* Y17 was performed with some modification by the Beijing Biomarker Technologies Corporation [22,64]. In order to increase marker specificity and efficiency, the SLAFs were compared with CS sequences [22] The SLAFs with less than 50% homology were used to design molecular markers. All these primers were synthesized by Shanghai Sangon Biotechnology Co., Ltd. (Shanghai, China).

These *Ae. biuncialis* specific markers were verified when their PCR bands were present in *Ae. biuncialis* Y17 but absent in common wheat CS. Those verified markers were further divided into U-specific and M-specific markers by amplifying *Ae. umbellulata* Y139 and *Ae. comosa* Y258. In addition, *Ae. biuncialis* specific markers (Appendix A) were used to characterize the 5M^b^ addition line. Here, *Ae. biuncialis*, *Ae. umbellulata* and *Ae. comosa* were used as positive controls, while CS as negative control.

PCR amplification was performed as described previously by Luan et al. [65]. The amplified products were separated by polyacrylamide gel electrophoresis (PAGE) with an acrylamide concentration of 8% and displayed by silver staining. The annealing temperatures of SLAF-based markers were 59 °C.

### 4.6. Agronomic Trait Determination of WA317

The wheat-*Ae. biuncialis* 5M^b^ addition line WA317 and its receptor parent CS were planted in a randomized complete block design with three replicates in Shijiazhuang, Hebei province of China. Thirty seeds of each line were evenly planted in six 2.25 m long rows, spaced 0.25 m apart. The agronomic traits were evaluated across 2017–2018 to 2018–2019 seasons.

At the physiology maturity stage, WA317 and CS were manually harvested. Measurement and counting were done on spike length, spikelet number per spike, grain number per spike and thousand-grain weight for 10 representative plants randomly selected in each plot. Statistical analyses were conducted using the Statistical Analysis System version 9.2 (SAS Institute Inc., Cary, NC, USA) and the *t*-test was used to test the difference of the agronomic traits between the addition line WA317and the receptor parent CS.

## Figures and Tables

**Figure 1 ijms-21-04053-f001:**
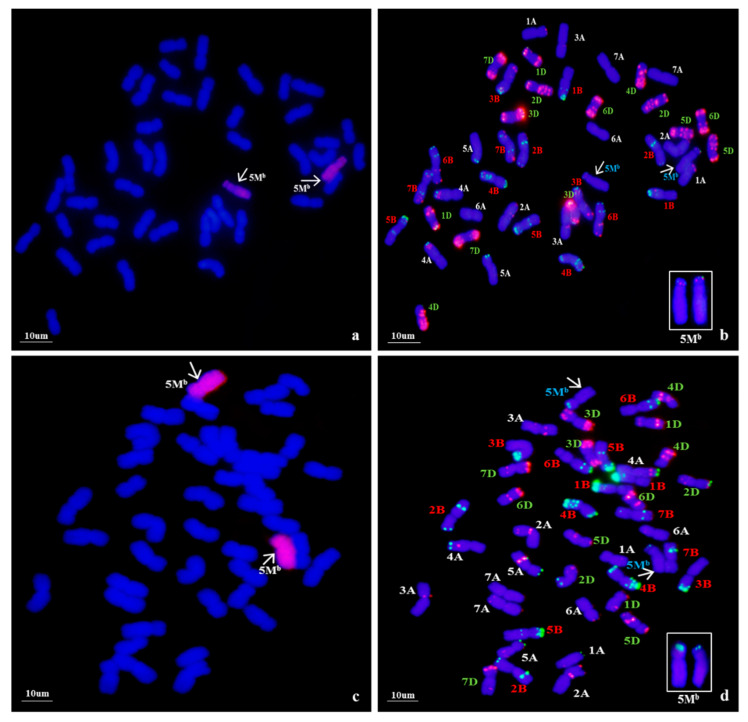
Genomic in situ hybridization (GISH) and non-denaturing fluorescence in situ hybridization (ND-FISH) detection of mitotic chromosomes in the root tips of wheat*-Ae. biuncialis* 5M^b^ addition line WA317. The arrows indicate the 5M^b^ additional chromosome. (**a**,**c**) GISH detection of WA317 using *Ae. biuncialis* genome DNA as probe; (**b**) ND-FISH pattern of the corresponding slide using probes Oligo-pAs1.0 (red) and OligopSc119.2 (green); (**d**) ND-FISH pattern of the corresponding slide using probes Oligo-pTa535 (red) and OligopSc119.2 (green).

**Figure 2 ijms-21-04053-f002:**
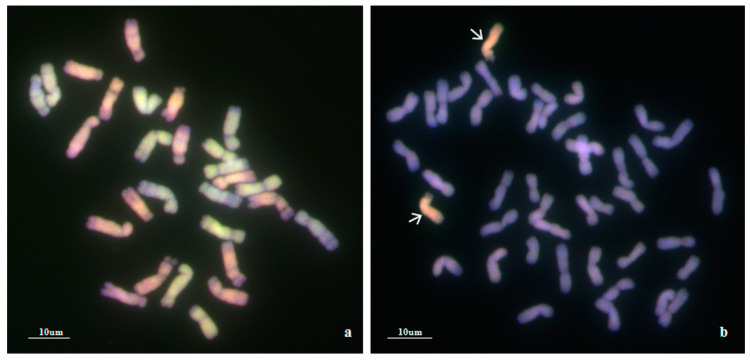
mcGISH detection of mitotic chromosomes in the root tips of *Ae. biuncialis* and wheat–*Ae. biuncialis* 5M^b^ addition line WA317 using *Ae. umbellulata* and *Ae. comosa* genomic DNA as probes. (**a**) mcGISH of *Ae. biuncialis*, U and M chromosomes are green and red, respectively; (**b**) mcGISH of WA317, the additional chromosomes are red and wheat chromosomes are blue. Arrows indicate the alien chromosomes from *Ae. biuncialis*.

**Figure 3 ijms-21-04053-f003:**
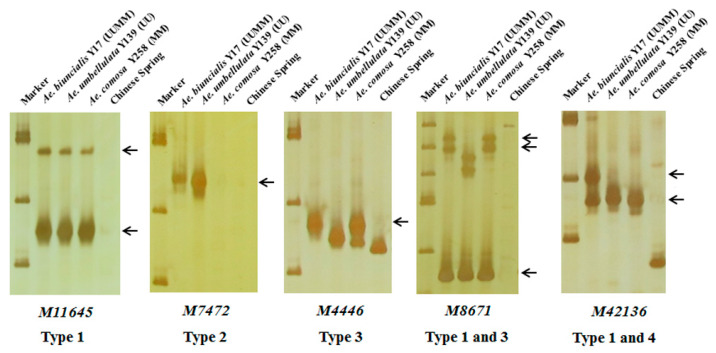
Amplification patterns of *Aegilops* accessions using *Ae. biuncialis* SLAF markers *M11645*, *M7472*, *M4446*, *M8671* and *M42136*. Type 1: the same amplified bands in *Ae. biuncialis* Y17 (UUMM), *Ae. umbellulata* Y139 (UU) and *Ae. comosa* Y258 (MM); Type 2: the U-specific amplified bands in Y17 and Y139; Type 3: the M-specific amplified bands in Y17 and Y258; Type 4: the specific amplified bands in Y17. Arrows indicate the specific bands of *Aegilops* accessions.

**Figure 4 ijms-21-04053-f004:**
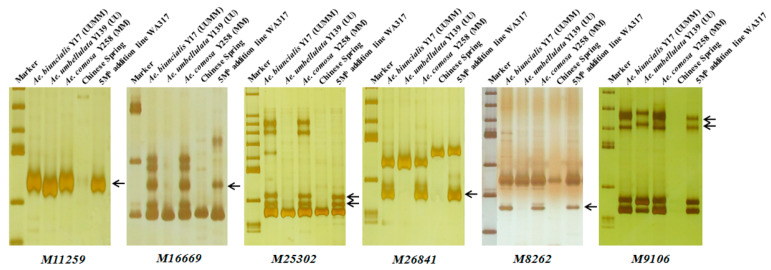
Polymerase chain reaction (PCR) amplification of six markers M11259, M16669, M25302, M26841, M8262 and M9106 for detection of 5M^b^ in WA317. These markers had the same M^b^-specific amplified bands in Y17, Y258 and WA317, which further confirmed that the alien chromosome in WA317 belonged to the M^b^ genome. Arrows indicate the M-specific bands in WA317.

**Figure 5 ijms-21-04053-f005:**
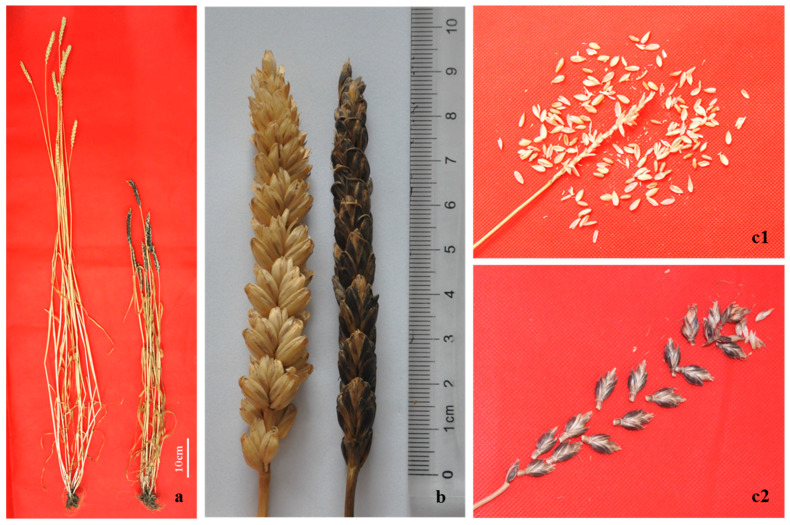
Morphological traits of wheat–*Ae. biuncialis* 5M^b^ addition line WA317 and its parent CS. (**a**) Plants of Chinese Spring (CS) and WA317 (left and right), (**b**) spikes of CS and WA317 (left and right), (**c**) Spikelets of CS (**c1**) and WA317 (**c2**) after threshing.

**Table 1 ijms-21-04053-t001:** Detection of *Ae. biuncialis* accessions and line WA317 using *Ae. biuncialis* specific-locus amplified fragment sequencing (SLAF) markers.

Type	*Ae. Biuncialis* Y17 (UUMM)	*Ae. Umbellulata* Y139 (UU)	*Ae. Comosa* Y258 (MM)	No. of Specific Markers	No. of WA317 Specific Markers
Type 1: U and M shared	+	+	+	172	37
Type 2: U-specific	+	+	-	6	0
Type 3: M-specific	+	-	+	102	6
Type 4: Y17-specific	+	-	-	18	2

Note: ‘+’, positive; ‘-’, negative.

**Table 2 ijms-21-04053-t002:** Agronomic traits of wheat-*Ae. biuncialis* 5M^b^ addition line WA317 and CS.

Trait	2017–2018 Season		2018–2019 Season	
	CS	WA317	CS	WA317
Plant height (cm)	114.60 ± 2.30	81.00 ± 3.35 **	120.25 ± 4.02	82.90 ± 5.02 **
Valid spikes per plant	11.83 ± 0.98	10.80 ± 0.84	12.16 ± 0.98	10.50 ± 1.22
Spike length	7.80 ± 0.34	7.50 ± 0.72	8.23 ± 0.65	8.00 ± 0.19
Spikelet number per spike	21.43 ± 1.51	19.20 ± 1.09	22.67 ± 1.37	20.80 ± 0.75
Grain number per spike	63.00 ± 4.64	42.60 ± 5.94 **	63.33 ± 4.63	44.00 ± 3.69 **
Thousand grain weight (g)	36.81 ± 1.56	30.77 ± 2.81 **	34.56 ± 0.42	26.05 ± 1.01 **

** indicates significant difference at *p* < 0.01 level.

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
