# Peer review of "Molecular Cytogenetic Identification of Wheat-Aegilops Biuncialis 5Mb Disomic Addition Line with Tenacious and Black Glumes"

_ijms, 2020, doi:10.3390/ijms21114053_

Round 1

Reviewer 1 Report

The authors present an interesting study aimed to identify a new wheat-Ae. biuncialis 5Mb addition line in the background of Chinese Spring, having obvious tenacious glumes, suggesting that Ae. biuncialis 5Mb chromosome carry genes functioning like gene Q located on wheat chromosome 5A. They used GISH analysis, non-denaturing FISH assay and SLAF-seq to assess the study hypothesis. The article it is an original contribution which has many good features including the experimental design and accuracy.

Author Response

Thank you for the thoughtful suggestions and the affirmation to our work.

Reviewer 2 Report

Review

The manuscript reports the molecular cytogenetic description of a wheat/Aegilops biuncialis 5Mb addition line and selection of Aegilops specific molecular markers. The work was carried out in an appropriate manner following established tools and methods.

Reading attentively the manuscript I would make the following remarks and suggestions:

Line 18 I would write elite genes originated from related species to wheat instead of „elite genes on alien chromosomes”

Line 21 I would use the term „had black color” instead of „significantly black”

Line 22 Please correct the misspelling of the word Triticeae

Line 28 Insert please „line” WA317

Line 29 Please correct the misspelling of the word Triticeae

Line 42 As not all the references mentioned on 42-44 lines describe wheat/alien addition lines (they are also partial amphiploids, introgression lines) please add the terms: partial amphiploids/introgression lines or replace the references with ones which describe wheat/alien addition lines.

Line 64 Please correct APTETALA to APETALA

Line 69-71 Please add some more recent references for genes controlling the glume color.

Line 79 Please correct Molnár team to Molnár-Láng team

Line 100 Can you provide the percentage of genetic stability for line WA317 disomic addition line (number of disomic plants/number of individuals check)?

Line 132 Table 1 Please make a clear division of rows indicating the four types of markers selected

Line 135 I suggest to enlarge a little bit the Figure 2 and to write on the top side of each column the genotype that belongs to. In this way, it would be easier to follow the bands specific to different genomes.

Line 137 Please explain the designation of pUC19 DNA/MspI (HpaII) as I did not find it in the text of the manuscript.

Figure 2 Please include gel electrophoresis patterns of markers specific to M genome which could support the findings of FISH analysis.

Line WA 317 complete with „line” WA 317

Line 178 Please correct the misspelling of Triticeae

Line 183 Correct the misspelling of Ae. umbellulata

Line 184 Correct the misspelling of Ae. geniculata

Line 264 The authors stated that: „The glume  (hardness and color) traits were also investigated”. Though data about glume hardiness were not included in the manuscript. Please add the results concerning this trait or correct the sentence.

Line 383 Correct the misspelling Molnárláng to Molnár-Láng

Line 394 Correct the misspelling Molnárláng to Molnár-Láng

The main questions about this manuscript are about the fact that molecular cytogenetic analysis results are not conclusive and the data of the selection of the molecular markers are not presented properly.

The authors described that the FISH analysis was carried out using the FISH karyotype established by Wang et al. (2013). The FISH hybridization signals detected by the Authors are different from those found in the referenced article. A karyotype of Ae. comosa M chromosomes is published by Molnár et al., (The Plant Journal, 2016). I agree that FISH hybridization patterns can be different for different accessions. In this case this should be mentioned in the Discussion part. I suggest to carry out a multicolour GISH analysis of the addition line WA 317, using the DNA originated from Ae. umbellulata (UU) and Ae. comosa (MM)as labelled probes, in this way the U or M genome origin of the Aegilops chromosome could be precisely defined. The Authors can also add a karyotype established by themselves (or refer to other publications) which supports their findings. Arm ratio can be as well a clue for identifying the additional chromosome but a similar arm ratio like 5M has, for example, the chromosome 2U.

The molecular marker analysis has the role to approve the results of molecular cytogenetic examinations. Checking the markers listed in Table S2 the results are included correctly in the table. By contrast, in the text (lines 142-143) this is not correctly stated:

„six markers (M8262, M16669, M26841, M25302, M9106 and M11259) amplified the same bands in Y17, Y139, and WA317, excluding Y258”.

Figure 2 presents the gel electrophoresis patterns of 6 markers analyzed. The last one would be an M specific marker (M16669). But checking the diagnostic bands this marker gives specific products for pUC19 DNA/MspI (HpaII), Ae. umbellulata and the addition line. Please check the figure, the legend and the text. To be more conclusive please add all the M specific marker photos even in an additional file.

I recommend a major revision of this manuscript.

Reviewer 3 Report

Molecular cytogenetic identification of 2 wheat-Aegilops biuncialis 5Mb disomic addition line 3 with tenacious and black glumes

Generally, the paper reads well, and the authors have clearly shown the introgression of an Aegilops biuncialis chromosome fragment into a hexaploid wheat cultivar Chinese Spring background using the GISH method.

However, I do have some concerns about this body of work before it can be published:

Introduction: Reads well- line 91- lowercase for ‘biuncialis’ Do double check elsewhere in text!

(1)    There are however, other issues that need to be addressed. The major concern is that only one plant line has been analysed. More lines will need to be analysed. In my own work using wheat transgenics either through RNAi or over-expression we would always use 3-5 lines when study a specific agronomic trait and the effect to ensure that what we are seeing, was based on the transgenic event and not some effect of the gene insertion.

(2)    Leading on from this more information on how the plant line was generated is needed. There is no information on how many crosses were done, how many plants were successfully generated, which was used as the female or male plant-were reciprocal crosses done, were the plants bagged to prevent any cross pollination, percentage success- this must be added. I refer the authors to the following:

BIOLOGIA PLANTARUM 54 (2): 259-264, 2010

259

GISH reveals different levels of meiotic pairing with wheat  for individual Aegilops biuncialis chromosomes

MOLNÁR and M. MOLNÁR-LÁNG

They should follow or use this paper as guideline on how to write up the ‘Plant Materials’ section.

(3)    Another concern is that the work does lack some important molecular analysis. Confirmation is based on scores of morphological traits and PCR based markers. Although the GISH method shows introgression of the 5Mb I feel the authors need to show specific gene expression either by RT-PCR or qPCR for specific genes which would confirm the specific transcript type and even some Western analysis to confirm that we are seeing is due to the introgression.

(4)    I also feel the other parent should be included the figures to show how different Aegilops biuncialis really is from Triticum aestivum and how the insertion of one chromosome has on the overall negative phenotypic and agronomic qualities for this line. This would add more gravitas as validation to the body of work carried out and highlight the importance of using these Aegilops species in gene introgression studies not only to introduce important agronomic traits but to decipher and understand gene function.

(5)    Sections 2.3 and 2.4 need perhaps to be restructured and merged into one section. Lead with the results confirming successful introgression and then discuss the development of the Aegilops specific markers? Not required but I feel it would read and flow better for the reader!

Round 2

Reviewer 2 Report

Reading the revised version of the manuscript I have the following remarks:

The major deficiency of the manuscript is an adequate and clear presentation of the results.

Unfortunately, the suggestions regarding the improvement of this section were taken into consideration only in a small compass by the Authors.

The multicolor GISH was not carried out, this would help to support the FISH and molecular markers analysis. The figure that was added to the Authors’ answers file is not mcGISH.

The authors claim that the hybridization patterns of 5Mb chromosomes are similar to the hybridization patterns of the chromosome 5M described in other publications: Wang et al., 2013; Molnár et al., 2016; Schneider et al. 2005; 2012.

Checking these publications, I’ve found that in all cases the 5M chromosomes examined by FISH give signs both on the short and long arm (e.g. pSc119.2 probe gives a terminal sign on the short arm of 5M and pAs1 gives interstitial signs on both short and long arm /Wang et. al., 2013/). These FISH probes give similar hybridization patterns like the oligo probes used in this manuscript. On figure 1 of the manuscript, I can see hybridization patterns only (terminal and subterminal) on the short arm of the chromosome named 5M. As I mentioned it is possible that different accessions give different hybridization patterns, but the FISH signals described in the manuscript are greatly different from those of the authors and publications mentioned as references.

If the FISH experiments don’t make possible a precise identification of the additional Aegilops chromosome, other (microsatellite) probes can be used. In order to improve in situ hybridization analysis I suggest reading of the following publications: DOI: 10.1556/AAgr.58.2010.3.1; DOI: 10.1186/s13039-014-0091-6.

A certain number of changes were made to Figure 2 but in my opinion, the figure doesn’t show/explain clearly which are M specific markers and which of them are suitable for identification of the chromosome named 5M in the WA 317 addition line. Figure legend is not complete and is not easily understandable.

English should be also improved.

Taking in consideration these remarks in my opinion this manuscript in the present form is not suitable for publication in IJMS.

Reviewer 3 Report

The authors have addressed most of my concerns but the main one still stands as this is publishing results on only 1 line.

Please do include photo of Aegilops parent please.
